# A Spontaneous Extracranial Internal Carotid Artery Dissection with Autosomal Dominant Polycystic Kidney Disease: A Case Report and Literature Review

**DOI:** 10.3390/medicina58050679

**Published:** 2022-05-20

**Authors:** Tsuyoshi Izumo, Yuka Ogawa, Ayaka Matsuo, Kazuaki Okamura, Ryotaro Takahira, Eisaku Sadakata, Michiharu Yoshida, Susumu Yamaguchi, Yohei Tateishi, Shiro Baba, Yoichi Morofuji, Takeshi Hiu, Takeo Anda, Takayuki Matsuo

**Affiliations:** 1Department of Neurosurgery, Graduate School of Biomedical Sciences, Nagasaki University, Nagasaki 852-8501, Japan; yukaxx0615@gmail.com (Y.O.); heygoo25@live.jp (A.M.); ocamoo@yahoo.co.jp (K.O.); myb020623@gmail.com (R.T.); e.sadakata@gmail.com (E.S.); michi511leo@yahoo.co.jp (M.Y.); ssmymgc@gmail.com (S.Y.); bb46.vader@gmail.com (S.B.); morofujiyoichi@gmail.com (Y.M.); thiu.nagasaki@gmail.com (T.H.); tanda-s@carol.ocn.nec.jp (T.A.); takayuki@nagasaki-u.ac.jp (T.M.); 2Department of Clinical Neuroscience and Neurology, Graduate School of Biomedical Sciences, Nagasaki University, Nagasaki 852-8501, Japan; ytate.com@gmail.com

**Keywords:** ADPKD, ICA dissection, plaque imaging

## Abstract

*Background and Objectives*: Non-cystic manifestation of autosomal dominant polycystic kidney disease (ADPKD) is an important risk factor for cerebral aneurysms. In this report, we describe a rare spontaneous internal carotid artery (ICA) dissection in a patient with ADPKD. *Observations**:* A 38-year-old woman with a history of ADPKD and acute myocardial infarction due to coronary artery dissection experienced severe spontaneous pain on the left side of her neck. Magnetic resonance imaging (MRI) revealed a severe left ICA stenosis localized at its origin. Carotid plaque MRI showed that the stenotic lesion was due to a subacute intramural hematoma. Close follow-up by an imaging study was performed under the diagnosis of spontaneous extracranial ICA dissection, and spontaneous regression of the intramural hematoma was observed uneventfully. *Conclusions*: When patients with a history of ADPKD present with severe neck pain, it is crucial to consider the possibility of a spontaneous ICA dissection. A carotid plaque MRI is beneficial in the differential diagnosis. Conservative management may benefit patients without ischemic symptoms.

## 1. Introduction

Autosomal dominant polycystic kidney disease (ADPKD) is one of the most common hereditary diseases characterized by bilateral renal cysts and liver cysts [1]. The non-cystic clinical manifestations of ADPKD include intracranial aneurysms, dissection of the thoracic aorta, and abnormalities of the cardiac valves [2,3]. Here, we report a case of rare spontaneous extracranial internal carotid artery (ICA) dissection in a patient with ADPKD, and her uneventful clinical course with conservative management.

## 2. Illustrative Case

A 38-year-old woman presented to our department with severe spontaneous pain on the left side of her neck. Her blood pressure was normal. Physical examination revealed no remarkable abnormality. Neurological examination revealed a score of 15 points (E4V5M6) on the Glasgow Coma Scale (GCS) without focal neurological deficit. She was previously diagnosed with ADPKD. She also had a history of acute myocardial infarction due to spontaneous coronary artery dissection shortly after delivering her first son 18 months prior and was relieved by conservative management. She had no history of hypertension, diabetes mellitus, or dyslipidemia. She also had no history of smoking and drinking. Her mother had been diagnosed with ADPKD. She had no family history of arterial dissection. Her carotid echo examination showed severe stenosis at the origin of the left ICA; the stenotic lesion was hypoechoic, and the peak systolic velocity increased to 332.1 cm/s at the distal ICA. Her echocardiographic findings were normal. Her abdominal computed tomography (CT) showed multiple cysts in the liver and bilateral kidneys (Figure 1A,B). Her initial neck magnetic resonance angiography (MRA) showed severe ICA stenosis at its origin (Figure 2A), and carotid plaque magnetic resonance imaging (MRI) showed a very high-intensity lesion causing severe stenosis on T1- and T2-weighted images (Figure 2B,C), which suggested subacute hematoma. Furthermore, the coronal section multiplanar reconstruction image showed a low signal intensity streak proximal to the stenosis, indicating an entry to the hematoma (Figure 2D). The findings from the imaging suggested that the stenosis at the origin of the left ICA was an intramural hematoma due to arterial dissection.

Although the patient had severe left ICA stenosis, she was without ischemic symptoms; therefore, we followed her closely with imaging studies without antiplatelet agents.

Her clinical course was uneventful. MRA performed three months after onset showed that the stenotic lesion of the left ICA had disappeared (Figure 3A), and plaque MRI confirmed that the intramural hematoma at the origin of the left ICA had disappeared (Figure 3B,C).

Written informed consent for treatment and publication was obtained from the patient.

## 3. Discussion

ADPKD is the most common hereditary renal disease [1]. It is characterized by the development of renal cysts leading to early renal function decline and end-stage renal failure [4]. ADPKD occurs in 1 out of 400–1000 individuals [4]. The most important non-cystic manifestation of ADPKD includes intracranial and other arterial aneurysms and, more rarely, dolichoectasias, dilatation of the aortic root, dissection of the thoracic aorta, and abnormalities of the cardiac valves [1,3].

Spontaneous extracranial ICA dissection associated with ADPKD is extremely rare, with only nine cases reported in the English literature to date, including our patient (Table 1) [5,6,7,8,9]. We analyzed these cases, which included four women and five men with a mean age of 41.1 years. For ordinary spontaneous extracranial ICA dissection, it is reported that the mean age of occurrence is approximately 45 years, and there appears to be a slight sex predisposition favoring males (53–57%) in population-based studies, which is similar to previous studies on ADPKD [10,11,12]. Neck pain as the presenting symptom was frequently observed in five out of nine cases, similar to that observed in ordinary spontaneous extracranial ICA dissection [12,13]. In the case of spontaneous extracranial ICA dissection, if cerebral ischemia or cranial nerve compression does not occur, neck pain can be the only symptom, as in our case. Thus, when a patient with ADPKD complains of neck pain, it is crucial to include cerebral ischemia and cranial nerve compression in the differential diagnosis and to perform adequate testing.

The detailed mechanism by which arterial dissection occurs in ADPKD patients remains unclear. ADPKD is a systemic disease resulting from mutations in either polycystic kidney disease-1 (PKD-1) or PKD-2 genes [1]. PKD-1 and PKD-2 mutations account for 85–95% and 5–15% of ADPKD cases, respectively [14]. PKD-1 encodes the polycystin-1 protein, and PKD-2 encodes polycystin-2 protein [15]. Recent studies have reported that polycystin is expressed in the epithelia and vascular smooth muscle cells and plays a crucial role in maintaining vascular integrity [14,16,17]. Therefore, in ADPKD cases, it can be assumed that mutations in these genes cause abnormalities in the maintenance of vascular homeostasis and cause arterial dissection.

The usefulness of vascular imaging has been reported in diagnosing spontaneous extracranial ICA dissection [18,19]. Mazzon et al. [18] reported that 168 cases of cervical artery dissection were examined by computed tomography angiography (CTA) and vascular MRI. The presence of an intramural hematoma was observed at a high rate in 113 cases (67.3%). On the other hand, dissection flap, which is a direct finding of artery dissection, was observed in 15 patients (8.9%) with a relatively low probability.

In our case, carotid echo examination showed a hypoechoic severe stenotic lesion at the left proximal ICA. The findings were similar to those of common atherosclerotic lesions and were non-specific. On the other hand, a high-resolution carotid plaque MRI showed a homogenous, high-intensity lesion causing severe stenosis both on T1- and T2-weighted images. These findings are consistent with a recent hemorrhage detected on high-resolution MRI [19]. Intra-plaque hemorrhage can also be observed in atherosclerotic carotid plaques. In such situations, however, similar findings are detected in only a part of the plaque [19,20]. Therefore, if the condition is homogenous throughout the lesion, as in this case, it is likely that the lesion would be an intramural hematoma due to arterial dissection. Furthermore, as in this case, the entry point to the dissection cavity and the dissection flap, which is a direct finding of dissection, might be confirmed; thus, if arterial dissection is suspected, the diagnosis should be verified using high-resolution MRI.

There is a lack of evidence regarding the optimal treatment of spontaneous extracranial ICA dissection. The recently published Cervical Artery Dissection in Stroke Study (CADISS) randomized clinical trial included 250 patients (including 224 patients with cerebral ischemia) within one week from onset to antiplatelet or anticoagulation therapy [21]. This study showed that there was no difference in ipsilateral stroke or ipsilateral transient ischemic attack between the antiplatelet (AP) arm (3.2%) and the anticoagulant (AC) arm (4.0%) (*p* = 0.58) during the 12-month follow-up. There was also no difference in the presence of residual narrowing or occlusion between those receiving AP (*n* = 56 of 92) versus those receiving AC (*n* = 53 of 89). Therefore, it is crucial to thoughtfully determine whether to use AP or AC for ischemic onset cervical artery dissections on a case-by-case basis.

Furthermore, there is no evidence of optimal treatment for spontaneous extracranial ICA dissection patients without ischemic symptoms. Published studies showed a similarly low risk of stroke and no evidence of an increased stroke rate in patients with ICA dissection [22]. Thus, except for hemodynamically compromised cases, it might be appropriate to closely follow patients via imaging studies without AP or AC in spontaneous extracranial ICA dissection cases without ischemic symptoms.

Spontaneous ICA dissection with ADPKD may have a benign prognosis similar to ordinary spontaneous extracranial ICA dissection. The reviewed cases and our cases were examined together, demonstrating that the prognosis was favorable in six cases, except for one patient with cerebral infarction, out of the seven total cases in which the prognosis was described (Table 1) [22]. Strunk et al. [5,6,7,8,9] reported that recurrent cervical artery dissection was rare in ordinary spontaneous cervical ICA dissection. On the other hand, as in our case, it is possible to develop arterial dissection at other sites; thus, careful observation of the clinical course is necessary for spontaneous ICA dissection in ADPKD patients [23].

## 4. Conclusions

We reported a rare case of spontaneous extracranial ICA dissection associated with ADPKD. If a patient with ADPKD complains of sudden neck pain, it is crucial to take a closer look considering the possibility of spontaneous extracranial ICA dissection. Cervical MRI is useful in extra cranial diagnostic imaging, especially in carotid plaque MRI, where an intramural hematoma can be identified as a lesion with more characteristic signal intensity depending on its stage. It is also helpful because high-definition images may show the entry point. The pathological condition is the same as that of ordinary spontaneous extracranial ICA dissection, and there is a high possibility of spontaneous resolution of the lesion. Therefore, conservative management based on close imaging follow-up is a practical option.

## Figures and Tables

**Figure 1 medicina-58-00679-f001:**
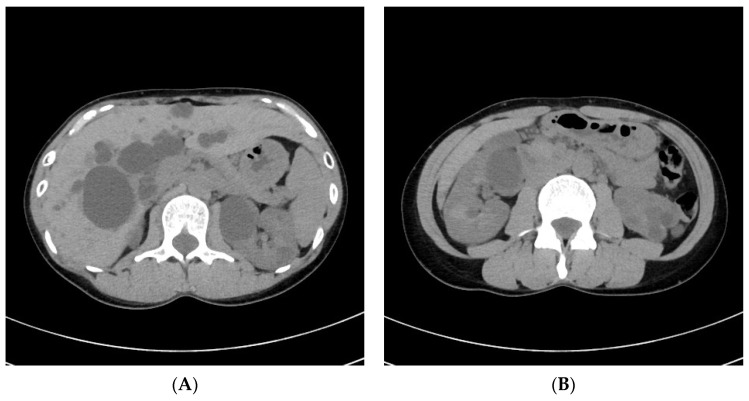
Abdominal CT (**A**,**B**) shows multiple hepatic and renal cysts.

**Figure 2 medicina-58-00679-f002:**
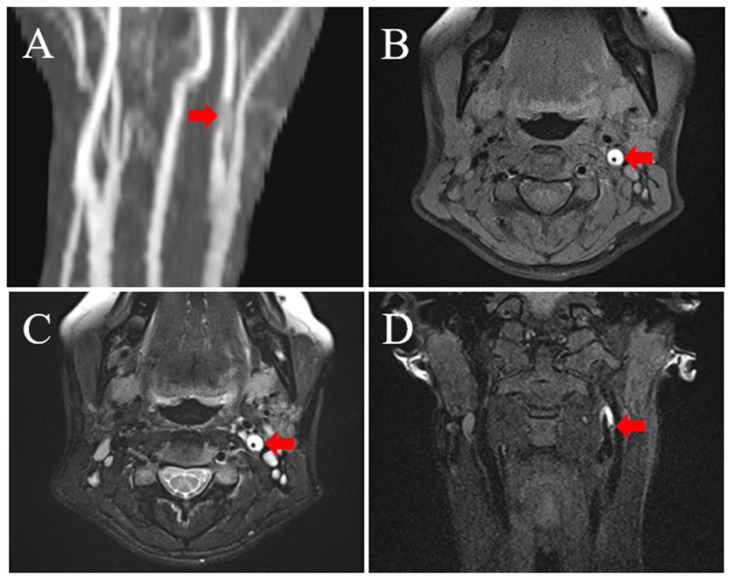
Initial neck MRA and MRI findings: (**A**) MRA shows severe stenosis at the origin of the left ICA and a faint hyperintensity lesion around it (red arrow). (**B**) MRI with axial fat-suppressed turbo spin-echo (FS-TSE) T1-weighted image shows hyperintensity lesion with an eccentric donut-like shape (red arrow). (**C**) MRI with axial FS-TSE T2-weighted image shows hyperintensity lesion with the same shape (red arrow). (**D**) MRI with coronal FS-TSE T1-weighted multiplanar reconstruction image shows a low signal intensity streak proximal to the stenosis (red arrow).

**Figure 3 medicina-58-00679-f003:**
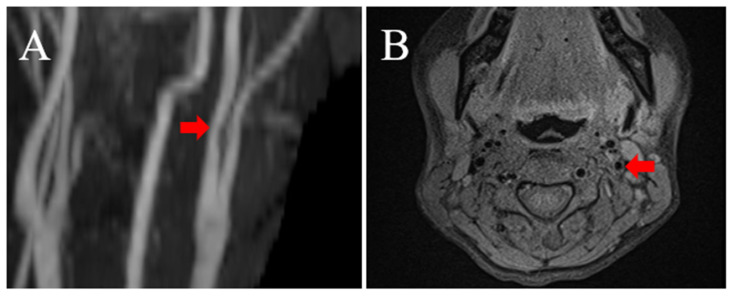
Follow-up neck MRA and MRI findings: (**A**) MRA three months after onset shows no stenotic lesion at the proximal ICA (red arrow). (**B**,**C**) MRI with axial FS-TSE T1-weighted (**B**) and T2-weighted (**C**) images show no intramural hematoma (red arrow).

**Table 1 medicina-58-00679-t001:** Summary of reported and present cases of spontaneous ICA dissection with ADPKD.

Author (Publication Year)	Sex/Age	Side	Presenting Symptoms or Signs	Treatment	Outcome (Causes)	F.Hx. of ADPKD
Bobrie G et al. (1998)	M/48	Lt.	Neck pain, rt. hemiparesis, aphasia	Anticoagulation	Favorable	No
	M/53	Rt.	Neck pain, lt. hemiplegia	N.A.	Sequelae (infarction)	No
	M/49	Rt.	Neck pain, Horner syndrome, lower CN palsy	Aspirin	Favorable	No
	F/34	Lt.	Asymptomatic	Aspirin	Favorable	N.A.
Roth C et al. (2013)	F/39	Bilat.	Headache, lt. hemiparesis	None	N.A.	Yes
Windpessl M et al. (2013)	F/35	Rt.	Headache, Horner syndrome	Heparin	Favorable	Yes
Kuroki T et al. (2014)	M/32	Lt.	Asymptomatic	N.A.	N.A.	Yes
Chen Z et al. (2019)	M/42	Rt.	Neck pain, rt. hypoglossal nerve palsy	Clopidogrel	Favorable	N.A.
Present case	F/38	Lt.	Neck pain	None	Favorable	Yes

ICA, internal carotid artery; ADPKD, autosomal dominant polycystic kidney disease; F.Hx., family history standard deviation; M, male; F, female; Lt., left; Rt., right; Bilat., bilateral; N.A., not applicable; CN, cranial nerve.

## Data Availability

Not applicable.

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
