# Peer review of "A Spontaneous Extracranial Internal Carotid Artery Dissection with Autosomal Dominant Polycystic Kidney Disease: A Case Report and Literature Review"

_medicina, 2022, doi:10.3390/medicina58050679_

Round 1

Reviewer 1 Report

The case report described by Izumo et al. provides a rare case of internal carotid artery dissection with ADPKD.

I have only few comments.

1. Please mention the family history of PKD and dissection of arteries.

2. Please mention the blood pressure and echocardiographic findings.

Author Response

Thank you for your detailed review.

We will provide a point-by-point response as below, and the manuscript has been revised along with them.

  1. Please mention the family history of PKD and dissection of arteries.

Her mother had been diagnosed with ADPKD. She had no family history of arterial dissection.

  1. Please mention the blood pressure and echocardiographic findings.

Her blood pressure was normal.

Her echocardiographic findings were normal.

Thank you again for your detailed consideration.

Reviewer 2 Report

Authors presented rare and interesting case report with appropriate discussion and  review of literature they can be commended for sharing interesting experience.

From the point of view of publication -  the presentation shows all appropriate parameters in high level.

Author Response

Thank you for your detailed review.

We will provide a point-by-point response as below, and the manuscript has been revised along with them.

From the point of view of publication -  the presentation shows all appropriate parameters in high level.

We want to express our sincere gratitude for your high evaluation of our manuscript.
